

# Development and validation of a nomogram to predict the mortality risk in elderly patients with ARF

Junnan Xu[1,*], Jie Weng[2,*], Jingwen Yang[3], Xuan Shi[3], Ruonan Hou[2], Xiaoming Zhou[2], Zhiliang Zhou[1], Zhiyi Wang[2,4] and Chan Chen[3]

[1] Department of Emergency Medicine, The Second Affiliated Hospital and Yuying Children's Hospital of Wenzhou Medical University, Wenzhou, China, China

[2] Department of General Practice, The Second Affiliated Hospital and Yuying Children's Hospital of Wenzhou Medical University, Wenzhou, China, China

[3] Department of Geriatric Medicine, The First Affiliated Hospital of Wenzhou Medical University, Wenzhou, China, China

[4] Center for Health Assessment, Wenzhou Medical University, Wenzhou, China, China

* These authors contributed equally to this work.

Corresponding authors
Zhiyi Wang, wzy1063@126.com
Chan Chen, chenchan99@126.com

## ABSTRACT

**Background**. Acute respiratory failure (ARF) is a life-threatening complication in elderly patients. We developed a nomogram model to explore the risk factors of prognosis and the short-term mortality in elderly patients with ARF.

**Methods**. A total of 759 patients from MIMIC-III database were categorized into the training set and 673 patients from our hospital were categorized into the validation set. Demographical, laboratory variables, SOFA score and APS-III score were collected within the first 24 h after the ICU admission. A 30-day follow-up was performed for all patients.

**Results**. Multivariate logistic regression analysis showed that the heart rate, respiratoryrate, systolic pressure, $SPO_2$, albumin and 24 h urine output were independent prognostic factors for 30-day mortality in ARF patients. A nomogram was established based on above independent prognostic factors. This nomogram had a C-index of 0.741 (95% CI [0.7058–0.7766]), and the C-index was 0.687 (95% CI [0.6458–0.7272]) in the validation set. The calibration curves both in training and validation set were close to the ideal model. The SOFA had a C-index of 0.653 and the APS-III had a C-index of 0.707 in predicting 30-day mortality.

**Conclusion**. Our nomogram performed better than APS-III and SOFA scores and should be useful as decision support on the prediction of mortality risk in elderly patients with ARF.

## INTRODUCTION

Acute respiratory failure is a common complication of critically ill patients. With an ageing population there is a growing prevalence of ARF. The incidence of ARF in the 65–84 age group was approximately twice that of the 55–64 age group and more than three times that of the young age group (*Flaatten et al., 2017*). In addition, ARF in elderly

patients is associated with a high mortality rate (*Behrendt, 2000*). The reasons included the heterogeneity and complexity of the elderly patient condition. Therefore, accurate assessment of the severity of ARF in the elderly patients is the key to reduce its mortality.

Academic research shows that clinical signs of ARF, including hypercapnia >45 mmHg, clearance of creatinine <50 ml minute-1, elevated NT-pro-B-type natriuretic peptide or B-type natriuretic peptide, were predictive of death (*Ray et al., 2006*). Although the risk factors are clear, there is still no consensus about the prognostic factors and evaluation system.

Some studies report that Simplified Acute Physiology Score-III (*Gannon et al., 2018*) and SOFA score (*Dziadzko et al., 2018*) can identify patients at high risk of death as reliably as the early warning score. with ARF. However, the sensitivity and specificity of these tools prediction are unsatisfactory. Moreover, it includes multiple indicators which are cumbersome to calculate. At present, there is a lack of clinical prediction tools of death for ARF in the elderly, therefore, it is urgent to find a risk stratification tool to predict mortality in elderly patients with ARF. The nomogram model can quantify, graph, and visualize Logistic regression results to achieve individualized prediction of disease risk. It has been successfully used in clinical diagnosis and prognostic assessment of various diseases (*Dong et al., 2018*; *Gilbride et al., 2018*).

In this study, we analyze the prognostic factors of elderly ARF patients, construct a nomogram model to predict survival, evaluate the risk of death of elderly patients, and then provide clinical help for early identification and intervention of high-risk patients to improve their prognosis.

## MATERIALS & METHODS

### Database and subjects

Subject data were retrieved from Medical Information Mart for Intensive Care III database version 1.4 (MIMIC-III v1.4) and the Second Affiliated Hospital and Yuying Children's Hospital of Wenzhou Medical University in China. The MIMIC-III is a clinical database comprising the information of 46,520 patients who were admitted to the ICU of Beth Israel Deaconess Medical Center (BIDMC) in Boston, MA, from 2001 to 2012 (*Johnson et al., 2016*). MIMIC-III is a large, freely accessible database for international researchers upon a use agreement (certification number: 31355221). The database was approved by the institutional review boards (IRB) of the Massachusetts Institute of Technology (MIT) and BIDMC, and consent was obtained for the original data collection (unidentified health information of patients was used); therefore, the informed consent was waived in our study. The MIT and BIDMC are responsible for waiving the informed consent.

### Participants

ARF was identified from ICD-9 code in the MIMIC-III database. For patients with multiple ICU admissions, we included only the first ICU admission. The primary outcome in this study was patients' 30-day mortality (died within 30 days after hospitalization).
## Data extraction

Demographical and laboratory variables were extracted from MIMIC-III database by Structure Query Language (SQL) at the first 24 h of ICU admission. We collected the following data: age, gender, vital signs, mechanical ventilation (including invasive and non-invasive), percutaneous oxygen saturation (SpO2), white blood cell (WBC), hemoglobin, platelet, albumin, bilirubin, blood urea nitrogen (BUN), lactate, activated partial thromboplastin time (APTT), prothrombine time (PT), 24 h urine output, sequential organ failure assessment (SOFA) score, and Acute Physiology Score III (APS-III). If the laboratory variables were examined more than once, the greatest severity value associated with the illness was used in our study. SOFA and APS-III scores were calculated within the first 24 h after the ICU admission. Patients with missing data, length of stay in ICU <24 h, and who were younger than 60 years old (*Lloyd-Sherlock et al., 2012*) were excluded from this study. Finally, 759 ARF patients were included in our analysis. All the scripts used to calculate the SOFA and APS-III scores were available from GitHub website (https://github.com/MIT-LCP/mimic-code/tree/master/concepts). Meanwhile, the same clinical data of ARF patients admitted to ICU in the Second Affiliated Hospital of Wenzhou Medical University from January 2010 to January 2020 were collected retrospectively. Inclusion criteria were length of stay in ICU ≥ 24 h, age ≥ 60 years old, and meeting the diagnostic criteria for acute respiratory failure (*Tierney et al., 2020*). Cases with missing data, advanced cancer, pregnancy, automatic discharge, and abandonment of treatment were excluded from the study. This study was approved by the Ethics Committee of the Second Affiliated Hospital of Wenzhou Medical University. IRB approval number: (2020) Ethical Approval No. 94.

## Statistics analysis

Continuous data are presented as mean ± standard deviation (SD) or median (IQR) according to the normal or non-normality distribution. Kolmogorov–Smirnov test was performed to determine normal distribution. Categorical variables were presented as frequency(proportion). Continuous data were compared using the Student $t$-test or Mann–Whitney U test and proportion variables were compared using chi-squared test or Fisher exact tests, as appropriate. For the development of the nomograms, the univariate and multivariable logistic regression were used to identify prognosis factors from the training data set. A nomogram was formulated based on the results of multivariable analysis, a final model selection was performed by a backward stepdown selection process with the Akaike information criterion (*Harrell Jr, KL & Mark, 1996*). The 'rms' package was used for nomogram and calibration curve (*Liu et al., 2020*). The accuracy of the nomogram to predict the 30-day mortality of ARF was quantified using the concordance index (C-index), which is equal to the area under the Receiver Operating Characteristic (ROC) curve and ranges from 0.5 to 1 (*Uno et al., 2011*). The difference of C-index was compared by DeLong's non-parametric approach (*DeLong, DeLong & Clarke-Pearson, 1988*). The calibration of the model is assessed by the calibration curves and determined using the Hosmer–Lemeshow goodness-of-fit test in the training set and validation set (*Grant, Collins & Nashef, 2018*). The 'rmda' package was performed for decision curve analysis (DCA) by

quantifying the net benefits to assess the clinical value of the model (*Vickers & Elkin, 2006*). We did the statistics analyses and figures production using R software (version 3.6.1). All statistics tests were two-sided, and *P* values <0.05 were considered statistically significant.

## RESULTS

### Patient characteristics

A total of 1,432 patients were included in this study. Patients from MIMIC-III database (759 cases) were categorized into the training set, the 30-day mortality in the training set were 38.6%. The 673 patients from our hospital were categorized into the validation set, the 30-day mortality was 40.5%. The patient characteristics and laboratory findings of the training and validation sets are shown in Table 1. There were no statistically significant differences between the training and validation set.

### Prognostic factors in the nomogram

Baseline demographic, laboratory variables, including SOFA and APS-III score for the prediction of 30-day mortality were determined using univariate logistic regression firstly. The heart rate, respiratory rate, systolic pressure, spo2, bilirubin, albumin, lactate, APTT, PT, BUN and 24 h urine output were prognostic factors of 30-day mortality in univariate logistic regression analysis. All above statistically significant prognostic factors were entered into the multivariable logistic regression for adjusting the confounding factors for 30-day mortality. The heart rate, respiratory rate, systolic pressure, $SPO_2$, albumin and 24 h urine output were independent prognostic factors for 30-day mortality (Table 2).

### Prognostic nomogram for 30-day mortality

We established this nomogram, elderly ARF mortality prediction nomogram (e-ARF-MPN), which incorporated the significant prognostic factors from the multivariable analysis (Fig. S1, Fig. 1). This e-ARF-MPN had a C-index of 0.741 (95% confidence interval [CI], 0.706–0.777) for predicting the 30-day mortality in ARF patients. Meanwhile, we developed two more nomograms according to SOFA and APS-III scores (Fig. S2). The SOFA nomogram had a C-index of 0.653 (95% CI [0.613–0.693]) and the APS-III nomogram had a C-index of 0.707 (95% CI [0.670–0.744]) (Supplemental Information). Our e-ARF-MPN had a better predictive power than both the SOFA ($P < 0.001$) and APS-III ($P = 0.039$) scores. For example, the calibration curve for e-ARF-MPN predicted 30-day mortality observed for in ARF patients in the training set was better than the SOFA and APS-III calibration curves (Fig. 2). Hosmer-Lemeshow test showed nonsignificant statistic (chi-square = 9.132, $P = 0.2197$) in the training set.

### External validation of the nomogram

The C-index of established e-ARF-MPN was 0.687 (95% CI [0.646–0.727]) for predicting 30-day mortality in the validation set. The C-index of APS-III and SOFA were 0.677 (95% CI [0.635–0.719]) and 0.613 (95% CI [0.569–0.657]), respectively. It has similar predictive power to the APS-III ($P > 0.05$), but significantly higher than the SOFA ($P < 0.001$). The calibration curves revealed adequate fit of the e-ARF-MPN and APS-III predicting the

**Table 1 Patient characteristics in training and validation set.**

| Variables | Training set (n = 759) | Validation set (n = 673) | P-value |
|---|---|---|---|
| **Age, median (IQR)** | 76 (68, 83) | 74 (66, 82) | 0.005 |
| **Gender, male, n (%)** | 395 (52.0) | 384 (57.1) | 0.064 |
| **Vital signs** | | | |
| Temperature, Median (IQR) | 37.6 (37.08, 38.25) | 37.44 (36.89, 38.06) | <0.001 |
| Heart rate, Median (IQR) (bmp) | 109 (95, 124.5) | 107 (92, 122) | 0.021 |
| Systolic pressure, Median (IQR) (mmHg) | 82 (72, 92) | 84 (73, 95) | 0.013 |
| Respiratory rate, Median (IQR) (per minute) | 28 (24, 33) | 28 (24, 33) | 0.748 |
| $SPO_2$, Median (IQR) (%) | 92 (88, 95) | 92 (88, 95) | 0.965 |
| **Laboratory findings** | | | |
| WBC, Median (IQR) ($10^9$/L) | 15.5 (10.6, 21.4) | 14 (10, 19.7) | 0.002 |
| Hemoglobin, Median (IQR) ($10^9$/L) | 9.6 (8.4, 11) | 9.8 (8.5, 11.1) | 0.279 |
| Platelet, Median (IQR) ($10^9$/L) | 181 (114.5, 250.5) | 178 (115, 250) | 0.846 |
| Albumin, Median (IQR) (g/dL) | 2.9 (2.4, 3.4) | 3 (2.5, 3.4) | <0.001 |
| Bilirubin, Median (IQR) (mg/dL) | 0.7 (0.4, 1.35) | 0.7 (0.4, 1.4) | 0.382 |
| Creatinine, Median (IQR) (mg/dL) | 1.5 (1, 2.4) | 1.5 (1, 2.6) | 0.323 |
| Glucose, Median (IQR) (mg/dL) | 107 (88, 133) | 109 (90, 133) | 0.293 |
| BUN, Median (IQR) (mg/dL) | 35 (23, 55) | 35 (22, 56) | 0.956 |
| APTT, Median (IQR) (s) | 36.3 (29, 54.95) | 35.7 (28.6, 54.6) | 0.203 |
| PT, Median (IQR) (s) | 15.4 (13.8, 18.5) | 15.6 (13.8, 20.3) | 0.062 |
| Lactate, Median (IQR) (mmol/L) | 2.7 (1.6, 5.15) | 2.3 (1.5, 4.1) | <0.001 |
| **Severity score** | | | |
| SOFA, Median (IQR) | 6 (4, 9) | 7 (4, 10) | 0.053 |
| APS-III, Median (IQR) | 58 (45, 76) | 60 (46, 81) | 0.159 |
| **24 h Urine output, Median (IQR) (ml)** | 1242 (676.5, 2112.5) | 1195 (625, 1950) | 0.201 |
| **Mechanical ventilation, n (%)** | 619 (81.6) | 545 (81.0) | 0.834 |
| **Length of stay, Median (IQR)** | 5.07 (2.59, 11.59) | 5.18 (2.55, 10.09) | 0.818 |
| **30-day mortality, n (%)** | 293 (38.6) | 273 (40.6) | 0.482 |

**Notes.**
SpO2, percutaneous oxygen saturation; WBC, white blood cell; BUN, blood urea nitrogen; APTT, activated partial thromboplastin time; PT, prothrombine time; SOFA, sequential organ failure assessment; APS-III, Acute Physiology Score III.

30-day mortality in the validation set, which is significantly better than the calibration curve for SOFA (Fig. 3). Hosmer-Lemeshow test showed nonsignificant statistic (chi-square = 10.086, $P = 0.2025$) in validation set.

## Decision Curve Analysis of the nomogram

The decision curve analysis (DCA) showed that this e-ARF-MPN had a large threshold probability range than the SOFA and APS-III scores. For example, whether mechanical ventilation should be used in a patient with respiratory failure. The decision curve analysis of our nomogram showed that the nomogram assisted- mechanical ventilation decision adds more net benefit to respiratory failure patients than either the treat-all-patients (all patients treated with mechanical ventilation) or treat-none-patients (no patients treated with mechanical ventilation) when the threshold probability is more than 8%. Compared to

**Table 2  The prognostic factors of 30-day mortality in univariate and multivariable logistic analyses.**

| Variables | Univariate | | | Multivariable | | |
|---|---|---|---|---|---|---|
| | OR | 95% CI | *P*-value | OR | 95% CI | *P*-value |
| Heart rate | | | | | | |
| ≤100 | reference | – | – | reference | – | – |
| 100~≤120 | 1.53 | 1.07–2.21 | 0.0218 | 1.25 | 0.84–1.89 | 0.269 |
| > 120 | 2.12 | 1.47–3.05 | <0.001 | 1.53 | 1.01–2.33 | 0.046 |
| Systolic pressure | | | | | | |
| ≥90 | reference | – | – | reference | – | – |
| ≥70~90 | 1.65 | 1.15–2.37 | 0.007 | 1.10 | 0.74-1.63 | 0.629 |
| ≥60~70 | 3.54 | 2.194–5.88 | <0.001 | 1.77 | 1.01–3.13 | 0.048 |
| < 60 | 7.84 | 4.42–14.38 | <0.001 | 3.68 | 1.91–7.28 | <0.001 |
| Respiratory rate | | | | | | |
| ≤20 | reference | – | – | reference | – | – |
| 20~≤25 | 1.18 | 0.61–2.41 | 0.619 | 1.31 | 0.64–2.79 | 0.475 |
| > 25 | 2.29 | 1.25–4.46 | 0.01 | 2.1 | 1.13–3.34 | 0.036 |
| SPO$_2$ | 0.96 | 0.94–0.97 | <0.001 | 0.97 | 0.96–0.99 | 0.005 |
| Albumin | 0.45 | 0.35–0.58 | <0.001 | 0.61 | 0.45–0.81 | <0.001 |
| Urine output | | | | | | |
| ≥1000 | reference | – | – | reference | – | – |
| ≥400~1000 | 2.52 | 1.78–3.58 | <0.001 | 2.11 | 1.44–3.09 | <0.001 |
| < 400 | 5.20- | 3.32–8.27 | <0.001 | 2.56 | 1.51–4.38 | <0.001 |

**Notes.**

SpO2,  percutaneous oxygen saturation.

the net benefit of SOFA- and APS-III-assisted decisions, at the same threshold probability, e-ARF-MPN showed higher net benefit. It unveiled the clinical utility of proposed e-ARF-MPN (Fig. 4).

# DISCUSSION

Nomogram is a visualization of regression analysis, which is widely used in clinical disease diagnosis and prognosis evaluation (*Callegaro et al., 2017*; *Chen et al., 2019*; *Wang et al., 2019*; *Zhou et al., 2019*). In this study, we developed and validated a *novel* e-ARF-MPN to predict the mortality risk among elderly patients with ARF. Our results show that this e-ARF-MPN is mainly based on *vital signs and laboratory examination*. The initial vital signs include heart rate, respiratory rate, systolic blood pressure, and SpO2, which were identified as independent predictors of mortality in elderly patients with ARF. With the increase of heart rate and respiratory rate, the risk of death increases. Furthermore, a decrease of systolic blood pressure and blood oxygen saturation will also increase the risk of death, both of which have a greater weight in the evaluation of short-term prognosis. Maintaining circulation stability and increasing blood oxygen saturation play an important role in reducing mortality of ARF in the elderly patients.
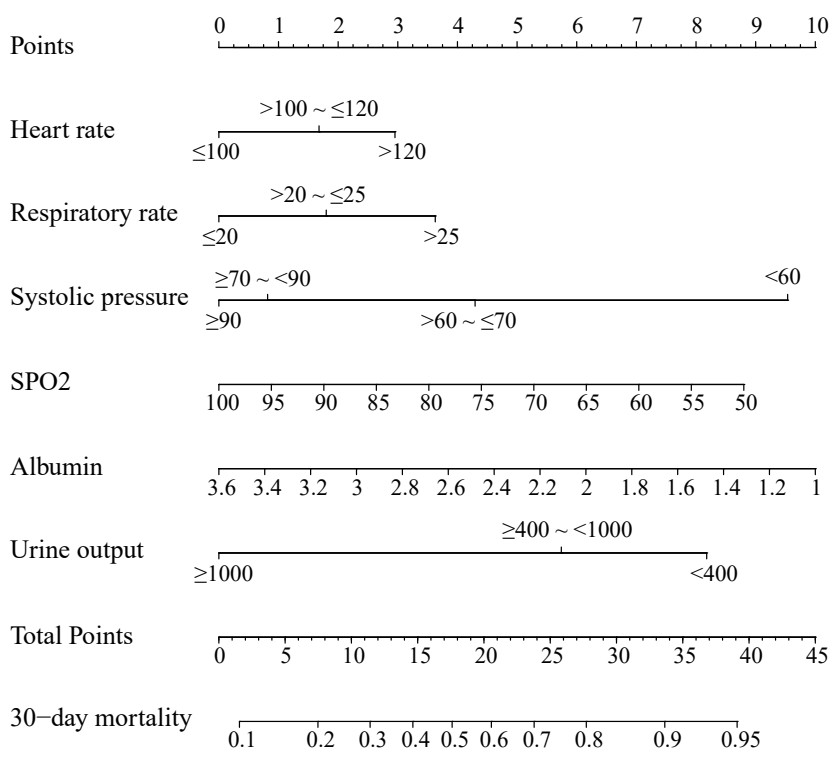

**Figure 1** **Nomogram to predicted 30-day mortality in ARF patients.** The nomogram was developed in the training set, with the heart rate, respiratory rate, systolic pressure, SPO2, albumin, and 24 h urine output incorporated.

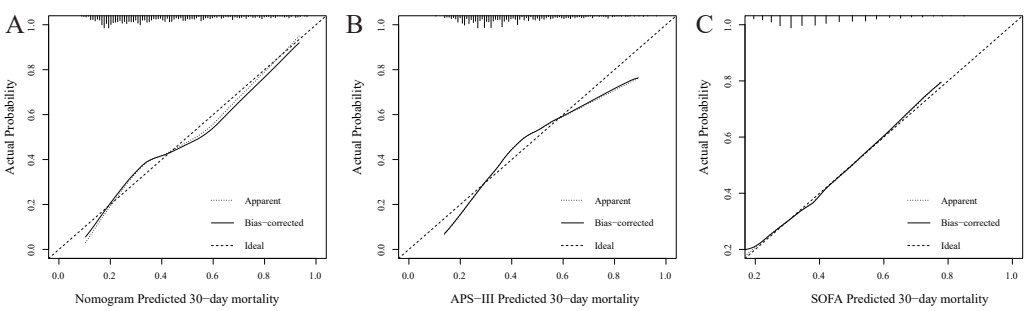

**Figure 2** **Calibration curves of the nomogram (A), APS-III (B) and SOFA (C) predicted 30-day mortality in training set.** Calibration curve represents the calibration of the nomogram, which shows the consistency between the predicted 30-day mortality and actual 30-day mortality of ARF patients. The $y$-axis represents the actual 30-day mortality, the $x$-axis represents the predicted 30-day mortality. The diagonal line represents a perfect prediction by an ideal model, and black solid line represents the prediction performance of the nomogram, of which a closer fit to the diagonal line means a better prediction.

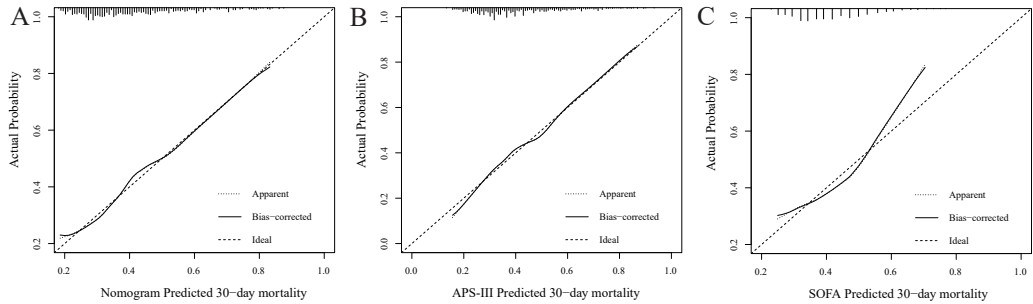

**Figure 3** **Calibration curves of the nomogram (A), APS-III (B) and SOFA (C) predicted 30-day mortality in validation set.** The $y$-axis represents the actual 30-day mortality; the $x$-axis represents the predicted 30-day mortality. The black solid line represents the prediction performance of the nomogram; the diagonal line represents an ideal nomogram model. The diagonal line represents a perfect prediction by an ideal model, and black solid line represents the prediction performance of the nomogram, of which a closer fit to the diagonal line means a better prediction.

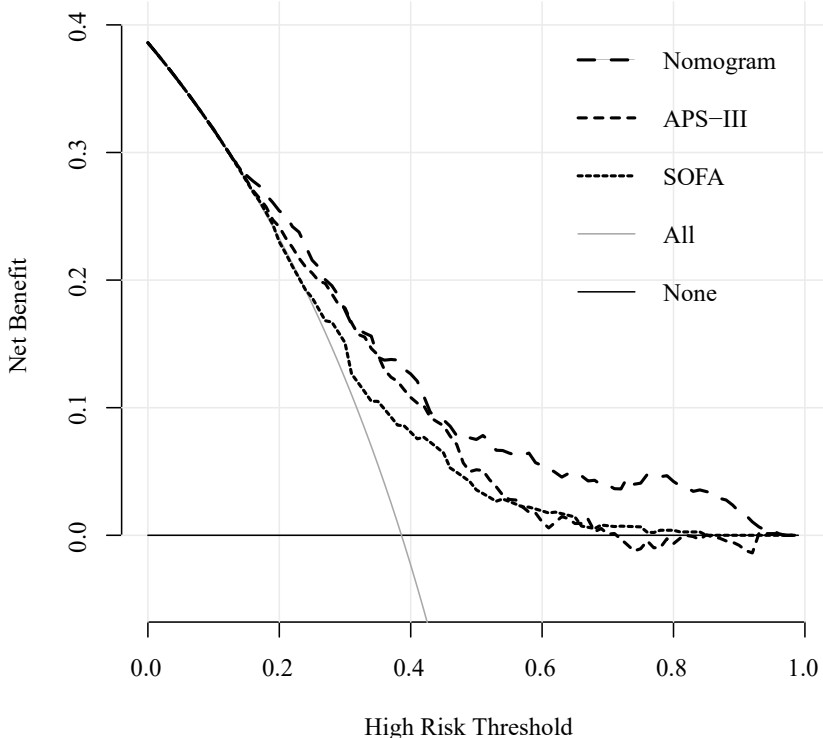

**Figure 4** **Decision curve for the training set cohort implicating the net benefit with respect to the use of the nomogram, APS-III, and SOFA score for predicting 30-day in ARF patients.** The $y$-axis represents the net benefit. The $x$-axis represents the threshold value. The red line represents the nomogram model, blue line represents the APS-III score and green line represents the SOFA score. The light grey line represents the assumption that all patients have the outcome (deceased). A thin black line represents the assumption that no patients have the outcome (deceased).

## Currently, urinary output and serum creatinine are used to evaluate kidney function

However, a study has shown that serum creatinine was an unreliable indicator of acute changes in renal function (*Li et al., 2016*). Our study also showed that urinary output was superior to serum creatinine in predicting short-term mortality of elderly patients with ARF. Although the assessment of AKI stage is not necessarily based on urine volume, the initial postoperative urine volume was considered an accurate predictor of delayed graft function (*Kim et al., 2019*). The reduction of urinary output can be attributed to insufficient blood flow to the kidneys, due to reduced blood volume and systolic pressure. Albumin, synthesized by the liver, is considered an important factor associated with malnutrition among patients. It tends to improve the microcirculatory performance which supports the maintenance of major organ functions (*Liu, Yang & Wei, 2017*). Thus, albumin was regarded as an important biomarker to evaluate the poor prognosis of hospitalized patients (*Akirov et al., 2017*). Our research also showed that the risk of death increased gradually with the decrease of plasma albumin. Therefore, plasma albumin may play an important role in predicting the mortality of elderly patients with ARF.

Finally, the e-ARF-MPN incorporates 6 items of heart rate, respiratory rate, systolic blood pressure, SpO2, urinary output, and plasma albumin. In order to prove the calibration of the nomogram, clinical data was collected from different institutions. As is well known, the internal validity associated with the explanation of the results, and the external validity related to the generalizability of the results (*Hong et al., 2019*; *Huebschmann, Leavitt & Glasgow, 2019*). Through the internal and external validation data set analysis, the calibration of our e-ARF-MPN has been proved to be highly consistent, which was more accurate than APS-III (B) and SOFA scores. At present, SOFA score has been widely used in assessment of critical diseases (*Gole, Srivastava & Neeraj, 2020*; *Mebazaa et al., 2018*), especially in the prognosis of multiple organ failure. When compared with APS-III and SOFA scores, nomogram developed in this study has fewer indicators but has better discrimination and calibration. This means that our nomogram may be popularized to predict the outcome of elderly patients with ARF.

However, evaluating the clinical usefulness of our nomogram depends on how much it benefits the patient, not just its popularization (*Huang et al., 2016*). DCA is a novel method that has been widely used in the evaluation of clinical research effectiveness (*Hijazi et al., 2016*; *Prabhu et al., 2019*; *Talluri & Shete, 2016*). It offers insight into clinical consequences on the basis of threshold probability, from which the net benefit could be derived (*Balachandran et al., 2015*). According to the DCA results, the application value of our nomogram is better than that of APS-III (B) and SOFA scores.

Our study has several limitations. First, our study was single center study and this e-ARF-MPN was only been validated in our hospital. We need to validate our e-ARF-MPN in broad external population. Second, our e-ARF-MPN is only applicable to the elderly ARF patients. Third, we reported 30-day all-cause mortality instead of ARF specific cause of death. Fourth, the time frame of our database collection i.e., from 2001 to 2012 is broad. Recent advance in the ARF treatment, i.e., noninvasive ventilation, high frequency oscillatory ventilation, and early intubation, might have a role in mortality.

## CONCLUSION

In conclusion, this study presents a novel nomogram that incorporates heart rate, respiratory rate, systolic blood pressure, SpO2, urinary output, and plasma albumin. It was better than both the APS-III (B) and SOFA scores; and could be useful as decision support on the prediction of mortality risk in elderly patients with ARF.

**Abbreviations**

| | |
|---|---|
| **ARF** | acute respiratory failure |
| **MIMIC-III v1.4** | Intensive Care III database version 1.4 |
| **SpO2** | Percutaneous oxygen saturation |
| **WBC** | white blood cell |
| **BUN** | blood urea nitrogen |
| **APTT** | activated partial thromboplastin time |
| **PT** | prothrombine time |
| **SOFA** | sequential organ failure assessment score |
| **APS-III** | Acute Physiology Score III |

### Funding

This study was supported by National Natural Science Foundation of China, No.81772054. Zhejiang Medicines Health Science and Technology Program, 2016KYB189. WenZhou Science and Technology Bureau, No. Y20170179 and Y20160114. The funders had no role in study design, data collection and analysis, decision to publish, or preparation of the manuscript.

### Grant Disclosures

The following grant information was disclosed by the authors:
National Natural Science Foundation of China: No.81772054.
Zhejiang Medicines Health Science and Technology Program: 2016KYB189.
WenZhou Science and Technology Bureau: Y20170179, Y20160114.

### Competing Interests

The authors declare there are no competing interests.

### Author Contributions

- Junnan Xu and Chan Chen conceived and designed the experiments, authored or reviewed drafts of the paper, and approved the final draft.
- Jie Weng conceived and designed the experiments, analyzed the data, authored or reviewed drafts of the paper, and approved the final draft.
- Jingwen Yang, Xuan Shi, Xiaoming Zhou and Zhiliang Zhou performed the experiments, prepared figures and/or tables, and approved the final draft.
- Ruonan Hou performed the experiments, prepared figures and/or tables, authored or reviewed drafts of the paper, and approved the final draft.
- Zhiyi Wang conceived and designed the experiments, analyzed the data, authored or reviewed drafts of the paper, and approved the final draft.

## Human Ethics

The following information was supplied relating to ethical approvals (i.e., approving body and any reference numbers):

The Training database was approved by the institutional review boards (IRB) of the Massachusetts Institute of Technology (MIT) and BIDMC. The validation data was approved by the Second Affiliated Hospital and Yuying Children's Hospital of Wenzhou Medical University granted Ethical approval [(2020) Ethical Approval No. 94]. Since this study is a retrospective study, it only collects the clinical data of patients with no treatment plans and will not bring risks to the patient's physiology. Therefore, the need for informed consent was waived in our study.

## Data Availability

Raw measurements are available in the Supplemental Files.

## Supplemental Information

Supplemental information for this article can be found online at http://dx.doi.org/10.7717/peerj.11016#supplemental-information.

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
