# Peer review of "Development and validation of a nomogram to predict the mortality risk in elderly patients with ARF"

_PeerJ, doi:10.7717/peerj.11016_

## Round 0.1 · original submission · Major Revisions

I have read your manuscript and the reviewers' comments, and I think your manuscript has high standards to be published in PeerJ. However, there are some issues which you must address in a revised version of the text. Please, see the reports below so as to have more information.

Best regards,
Dr Palazón-Bru (academic editor for PeerJ)

Reviewer 1 ·

Basic reporting

Dear authors, I enjoyed reading your manuscript. I believe you report on a relevant topic and think the proposed nomogram is potential useful in clinical settings. However, I believe some points should be further clarified.

The report is well structured.

The professional use of the English language should be checked by a native English speaker. An example can be found in line 44.

Figure 3: The Figures of the calibration curves are blurred (in my review version, so please check)

Experimental design

Line 55 Research question:
Please clarify the precise survival (30d) you assess with your study.

Validity of the findings

Line 68:
Please add name the name of the institution responsible for waiving the informed consent.

Line 69 following:
Please report the methods you used for the recruitment of participants in the validation set within the appropriate method section. Also please report the responsible ethical committee.

Line 69 following:
Please state how the predictors and the outcome were assessed within the validation data set.

Additional comments

Line 83:
Please add the reference for the WHO statement

Line 95:
Please add the reference to the “rms” package.

Line 97:
Please use the full name "concordance index" for the first time.
Please provide a guide how you interpreted the C-index.

Line 97:
Please add the guide, which was used to assess the calibration curves.

Line 98:
Please provide an interpretation guide and the package or statistical software application you used for the decision curve analysis.
It would be helpful if you could reference a method paper, which supports or recommends both statistical procedures (C-index and DCA) for the evaluation of prediction models.

Results section:
Line 108 or elsewhere:
Please report the amount of missing data for the predictors and the outcome.

Line 108 or elsewhere:
Please indicate if any participants were excluded from the analysis?

Line 110-117 or Table 2
Please report your regression models in more detail. There exist several guidelines to report regression analyses. For example, Lang T, Altman D. Statistical Analyses and Methods in the Published Literature: the SAMPL Guidelines.
Link:
https://www.equator-network.org/reporting-guidelines/sampl/

Line 119 and Figure 2:
The nomogram is the key of this paper. I believe it should be reported in more detail. How are the points awarded to the different predictors? Is a summary score build? Or is each item scored individually? Please expand your reporting to allow the replication of the Nomogram in clinical and research settings.

Line 122:
I suggest adding the reference to both nomograms within brackets (such as additional information 1 and 2).
Similar to the comment above (comment line 119). Could you please provide a short description how the nomograms can be used? For example, it looks like a SOFA score of 22 points equals 10 Total points. But what is the difference between “Total Points” and “Points”?

Line 125-127
As mentioned above (comment line 97): please provide more information on how you categorised the calibration curves into “good” or “better”.

Line 133:
If you refer to significantly, this sounds like you performed a statistical test to compare the calibration curves?

Line 137-140:
Your statement regarding the clinical utility is relatively vague. Could you clarify what aspects you consider under the term “clinical utility”.

Reviewer 2 ·

Basic reporting

The tables and figures are well presented. However, the p-values should be listed in Table1.

Experimental design

1. It is suggested that the authors give a name of their final model instead of calling it nomogram, otherwise it can be easily confused with the plot.
2. In the statistical analysis section, the authors wrote ‘The Student t-test, Mann-Whitney U test, chi-squared test, or Fisher’s exact test was performed where appropriate. ’ It is suggested to describe this in a neat way.
3. The criteria of candidate variables entering into multivariate model is not clear in the statistical analysis section. It wasn't clear on the meaning of ‘showed statistics relationship’.
4. Was there any variable selection method implemented? In Table2, it seems that there are 5 out of 11 variables are insignificant. A variable selection, either a stepwise or purposeful variable selection process, should be implemented in order to get the final model with the least amount of variables.
5. It wasn’t clear if the authors did any transformation to the laboratory values in the model, or checked the linearity between variables.
6. What are the missing percentage of the candidate variables?

Validity of the findings

1. In the section of ‘Prognostic nomogram for 30-day mortality’, the authors calculated the C statistics for different models, however, the 95% CIs of the APS-III model and the proposed nomogram model have some overlaps, the authors should be cautious about stating that ‘our nomogram performed a better predictive power than APS-III score model’. A test to compare the C statistics could be added to strengthen this conclusion. Same for the validation cohort.
2. In figure 2, it is hard to tell from the plot that ‘the nomogram was better than the calibration curves for SOFA and APS-III’. The authors should report the Hosmer-Lemeshow GOD chi-square statistics and its p-value to strengthen the conclusion. Same for the validation cohort.
3. It would be interesting to see the subgroup results in different risk factor groups or different time periods.

Additional comments

This is an interesting study to identify the common prognostic risk factors in predicting short term mortality in elderly patients with ARF. The authors constructed the logistic regression models to predict the risk of 30-day mortality and compared the final model with two established risk scores. Although the problem and the clinical importance of this work is well stated, some minor points need to be addressed and further discussed to make it suitable for publication.

---

## Round 0.2 · Minor Revisions

Still pending some minor modifications suggested by one of the reviewers.

Reviewer 1 ·

Basic reporting

ok

Experimental design

ok

Validity of the findings

ok

Additional comments

Dear authors and editors,

Thank you for giving me the opportunity to review the resubmitted manuscript. The authors have responded to my comments and I propose to accept the paper.

Reviewer 2 ·

Basic reporting

no comment

Experimental design

no comment

Validity of the findings

Minor suggestions:
1. In the statistical section, 'The frequency (proportion) for categorical variables. ', this should be revised as 'Categorical variables were presented as frequency(proportion).'
2. The name of test used for comparing C-indices should be included. The corresponding reference should also be inserted.
3. Table2 should only include the final subset of variables for multivariable results, not all the candidate variables.

Additional comments

Thanks for the feedback. The manuscript looks good to me.

---

## Round 0.3 · accepted · Accept

All the reviewers' concerns have been correctly addressed.

Reviewer 2 ·

Basic reporting

no comment

Experimental design

no comment

Validity of the findings

no comment

Additional comments

Thank you for addressing my questions. The manuscript looks good to me and no further questions from my end.